# Basics of faculty-to-faculty mentoring: A process to identify support and challenges

**Tabassum Zehra** [1] *, **Muhammed Tariq**[1], **Rehana Rehman**[2], **Rukhsana W. Zuberi**[1]

**1** Department for Educational Development, Aga Khan University, Karachi, Pakistan, **2** Department for Basics & Biological Sciences, Aga Khan University, Karachi, Pakistan

These authors contributed equally to this work.

* tabassum.zehra@aku.edu

**Data Availability Statement:** All relevant data are within the paper and its Supporting Information files. Minimal data set is uploaded on BioStudies https://www.ebi.ac.uk/biostudies/submissions/files.

## Abstract

Faculty mentoring is relatively new in health sciences literature. Faculty mentors are considered to have multiple roles including being a supervisor, teacher, or a coach. Little or no attention to formal mentoring of faculty allows faculty to seek informal mentoring, creating a risk of unexpected outcome. There is dearth of literature with respect to formal mentoring programs from the subcontinent. Although, informal faculty mentoring has been in place but there is no standard faculty mentorship model to be followed at Aga Khan University Medical College (AKU-MC). An observational study was conducted in September 2021 at AKU MC with convenient sampling to share the perceptions of the AKU-MC faculty mentors in a faculty mentorship workshop so that further advanced level faculty development workshops could be planned in this area. Twenty-two faculty mentors participated to share their perspectives on the responsibilities of faculty mentor, faculty mentee and the institution to see faculty grow and to have a sustainable mentorship programme. Challenges faced by the faculty mentors during the process of mentorship were also discussed. Majority of the participants emphasized on the role of the faculty mentor to be supportive, guiding, reflective and formative (respond to the emotional needs, encourage, effective communication, know own limitations, observe, provide feedback). Faculty mentor role modeling, ability to maintain confidentiality, creating and maintaining mentor-mentee relationship, availability of framework of formal mentoring in an academic institution and opportunities within the academic setting to learn mentorship were the main challenges of being a faculty mentor. The process provided valuable training and education to the faculty for the development and strengthening of formal mentoring program. Faculty recommended that institutions should provide the opportunity for development of junior faculty mentors by organizing capacity building activities.

## Introduction

Capacity building programs in medical education were initiated formally in the late twentieth century [1] and is seen as a common practice in Lower Middle Income Countries [2]. Faculty mentoring is relatively new in health sciences literature. In medical education faculty

**Funding:** The author(s) received no specific funding for this work.

**Competing interests:** The authors have declared that no competing interests exist.

mentoring is defined as "a reciprocal learning relationship characterized by trust, respect, and commitment in which a mentor supports the professional and personal development of another (the mentee) by sharing his or her life experiences, influence, and expertise" [3].

Academic institutions across the globe consider mentoring as a tool for faculty development. Literature reports limited evidence from the subcontinent with respect to formal faculty mentoring programs in the academic institutions and faculty perspectives to strengthen this area Furthermore, third world countries lack structured faculty mentoring programs [4] which is considered to be the need of the day.

Faculty mentoring in medical education can help personal and professional development of faculty in teaching, research and career planning [5, 6]. Faculty mentors are considered to have multiple roles including being a supervisor, teacher, or a coach. Mentoring requires building a mentor-mentee relationship to support and guide the mentee to achieve wide array of objectives including emotional support, personal and professional development. Faculty mentoring needs & relationships change over time (evolving interests, needs, time commitments etc).This is unlike teaching which focuses on attainment of specific outcomes [7]. Little or no attention to formal mentoring of faculty permits faculty to seek informal mentoring, creating a risk of unexpected outcome [8]. Faculty lacks the required knowledge and understanding of their role as mentors irrespective of their level. Faculty display less positive attitude towards mentorship, and have ineffective involvement in the mentoring process and faculty development in this area [9].

To initiate any mentoring programme, the basic plan of delivery and conduct should be devised by the medical educators in collaboration with senior faculty members. This enables to identify the needs of the junior faculty members in terms of the support required [4] and the challenges faced by the faculty members in the mentoring process.

## Rationale

Although, informal faculty mentoring (apprenticeship) has been in place but there is no standard faculty mentorship model to be followed at AKU-MC [1] yet the importance of faculty mentoring was highlighted at multiple fora within the university. Therefore, Aga Khan University Medical (AKU-MC) College initiated formal faculty mentorship program in 2019 with the aim to provide a conducive learning environment for capacity building of the faculty mentors at AKU-MC. The rationale for the initiation of this program was to attach newly appointed junior faculty with a senior faculty for an understanding of AKU's vision, mission and organizational culture for a smooth transition into their work and practice. Literature supports the role of workshops for training of mentors [10]. A series of workshops for faculty at AKU-MC were planned from the platform of Faculty Mentorship Forum in collaboration with Department for Educational Development to explore the perception of mentors and enhance their mentorship capability.

In this context, the first workshop was conducted with the objectives to 1) identify the needs of AKU MC faculty mentors, mentee & institution, ii) explore faculty perspectives on responsibilities & expectations of faculty mentor, faculty mentee, and the institution iii) identify challenges faced with respect to being a faculty mentor iv) identify the scope of mentor-mentee relationship. The perceptions of the AKU-MC faculty mentors would then help better understanding of the Faculty Mentorship Forum to base further advanced level faculty development workshops in this area.

Research Question:

What are the perceptions of faculty mentors of Faculty Mentorship Forum about mentorship at AKU-MC?

## Methodology

An observational study was conducted in September 2021 at AKU MC with convenient sampling. Twenty-two faculty members registered with Faculty Development Forum as faculty mentors attended the virtual three-hour workshop with their consent taken by the Faculty Mentorship Forum at the time of nomination via email. Faculty from outside AKU-MC were not included. Medical educators from Department for Educational Development which included senior faculty members, having an experience of being faculty mentors, facilitated the workshop. The study did not require ethical approval since the aim was to report the process and faculty perspectives. Verbal consent was sought by facilitator at the time of the conduct of the workshop from the participants to report the process and their perspectives. Verbal consent was sought and recorded on ZOOM for recording of the proceedings and dissemination of the results. The workshop started with seeking expectations of the faculty mentors with respect to the workshop. The workshop participants were asked to share their perspectives on the responsibilities of faculty mentor, faculty mentee and the institution to see faculty grow and to have a sustainable mentorship programme. An interactive faculty presentation was made on different models available of faculty mentorship along with the roles and responsibilities of each stake holder defined in literature. The participants were divided randomly in three different virtual small groups on ZOOM for three scenario-based discussions on the scope of mentoring to address benefits of mentorship process to the faculty mentor and faculty mentee. Each group was then supposed to role play the case in the larger group on ZOOM. Three scenarios were developed around i) Personal Development of faculty mentee ii) Professional Development of faculty mentee iii) Faculty mentor promotions/appointments/ salary. Scenario one (Personal Development) focused on effective communication skills, coping with AKU culture, exploring faculty mentee potential (leadership etc) and organization and management. Scenario two (Professional Development) focused on matching individual faculty mentee needs with gaps at departmental level and institutional needs at AKU. Scenario three (Promotions/Appointments/Salary) focused on promotion of a junior faculty mentee (Table 1). Responses were recorded and notes were taken by the facilitators. Member checking was done simultaneously.

**Table 1. Scenarios given to the participants.**

**Scenario 1: Personal Development**
An Assistant Professor is assigned to you as his/her mentor. The faculty mentee is concerned about his/her personal growth. The faculty mentee has come to you to discuss and seek advice on this matter. Create a role play to demonstrate a particular situation of your faculty mentee's personal development through
- Inculcating purpose and personal goals
- Sensitizing cultural growth / awareness
- Intellectual development
- Exploring potential (leadership etc.)

**Scenario 2: Professional Development**
A Senior Instructor at Department of Surgery/ Medicine is assigned to you as his/her mentor. The faculty is concerned about his/her career growth and professional development. Create a role play to demonstrate a particular situation in which to mentor your mentees professional development through
- Matching individual needs with gaps at departmental level
- Matching individual needs with institutional needs
- Choosing an area/clinical discipline which is missing at AKU

**Scenario 3: Promotions**
An Assistant Professor is assigned to you as his/her mentor. The faculty has been working at the same level for six years. As a mentor you are interested in your mentees promotion to Associate Professor. The faculty portfolio was submitted for promotion, but the application got declined. Create a role play to demonstrate a particular situation on mentoring the faculty for his/her promotion.

**Table 2. Participant expectations from the workshop: On a scale of three (03) where 1 = Agree, 2 = Don't Agree and 3 = No response, the majority agreed to the overall expectations.**

| | | | Mean | SD |
|---|---|---|---|---|
| 1. | Technicalities of being a mentor/experience of others (peer learning) | | | |
| 2. | Interaction/communication between mentor and a mentee | | 1.27 | 0.703 |
| 3. | Meaningful contribution in development of mentee (communication, process) | | 1.45 | 0.858 |
| 4. | Scientific tools available | | 1.36 | 0.790 |
| 5. | Formal vs informal mentoring | | 1.45 | 0.858 |
| 6. | Mentorship process (responsibilities, relationship etc) | | 1.45 | 0.858 |
| 7. | Mentoring support available and allied factors (institutional) | | 1.27 | 0.703 |
| 8. | Responsibilities of a mentor | | 1.73 | 0.985 |
| 9. | Basics of mentoring | | 1.09 | 0.426 |
| 10. | Strategies unexplored as a mentor/improve self | | 1.82 | 1.006 |
| 11. | Learn new techniques | | 1.36 | 0.790 |
| 12. | Impact of mentorship | | 1.45 | 0.858 |
| 13. | Way forward for formal learning | | 1.27 | 0.703 |

## Results

Twenty-two faculty (n = 22) attended the workshop. Participants expected the workshop to cover areas related to the process, relationship, personal development and roles of faculty mentors (Table 2) which was covered during interactive discussion and role plays.

Participants responded on different aspects of the roles of faculty mentor, faculty mentee and role of the institution (Table 3).

Majority of the participants emphasized on the role of the faculty mentor to be supportive, guiding, reflective and formative (respond to the emotional needs, encourage, effective communication, know own limitations, observe, provide feedback)

Participants responded that faculty mentor role modeling, ability to maintain confidentiality, creating and maintaining mentor-mentee relationship, availability of framework of formal mentoring in an academic institution and opportunities within the academic setting to learn mentorship were the main challenges of being a faculty mentor. Other challenges included honest assessment by the faculty mentor and opportunity for formal Faculty Mentorship Programme within the institution (Table 4).

**Table 3. Role of faculty mentor, faculty mentee, and institution.**

| Role of Faculty Mentor | Role of Faculty Mentee | Role of the Institution |
|---|---|---|
| supports the mentee (supportive) | sets the goals for the relationship | **creating a culture of mentorship from top to bottom** |
| helps developing a career path | seeks guidance | provide a **formal** mentorship training program |
| flourishes the capabilities of the faculty mentee | **recognizes and accept** faculty as a mentor | Provide adequate **protected time** |
| demonstrates interpersonal & **communication skills** | have **open communication**, | Provide resources |
| knows **own limitations** | **accepts accountability** | appreciate and **reward** effective mentorship |
| responds to the **emotional needs** of the mentee | maintain relationship, (rapport building), | provide a **framework for ground rules** surrounding mentor-mentee relationship |
| observes the actual progress | focused on career growth, | |
| give constructive feedback | seeks feedback and develops an **action plan.** | |

**Table 4. Challenges of being a mentor.**

| |
|---|
| Creating balance between service, teaching & research** |
| Optimize meaningful relationship** |
| Formalize Faculty Mentorship Programme |
| Framework required in academic setting** |
| Ability to absorb anything** |
| Honest assessment |
| Trust & confidentiality*** |
| Role modeling** |
| Allied factors |
| Opportunities to learn mentorship*** |

*Frequency of responses

## Discussion

Literature reports that a less experienced faculty mentor would benefit from a formal ongoing mentorship program opportunities available within the academic institution [7, 11]. The benefits for the junior faculty mentor would include personal and professional development of required set of skills for mentoring, effective communication skills, and improved subject knowledge expertise [12].

Our outcomes of faculty mentorship workshop are similar to series of mentorship training workshop which were endorsed by mentors for development of mentoring in low middle income countries [13].The involvement and focus of the participant with respect to the process, mentor-mentee relationship, personal development, and roles of faculty mentors was covered during interactive discussion and role plays. This supported the exchange of ideas in the development of mentor-mentee relationship, faculty mentor communication and leadership skills. They were able to define faculty mentoring needs, and discuss recruitment methods and institutional policies for reward and recognition which corroborates with literature [14].

The facilitators were able to cover almost all the expectations of the participants through presentation, interactive discussions, brainstorming and role plays. One of the participants expected to understand the impact of mentorship workshop, which was beyond the scope of the session.

The participant's responses with respect to the role of faculty mentor, faculty mentee and the institution were categorized either to be supportive, guiding, reflective and formative. (responds to the emotional needs, encourage, effective communication, know own limitations, observe, provide feedback etc). This is line with another study which reports faculty mentors provide guidance, support, advise and coach faculty mentee in their development [5]. However, the participants did not mention the role of faculty mentors in providing guidance to faculty mentee to develop leadership skills and networks through socialization of their profession. Similarly, the role of faculty mentee in developing portfolio and personal satisfaction did not come across the discussion. Also, the institutional responsibility of providing opportunities for social mobility was not discussed [7, 15].

Faculty mentors get benefit of the institutional mentorship programs. However, lack of adequate capacity building of faculty mentors without any appropriate model, may give rise to certain challenges that the faculty mentors face during the process, especially when the faculty mentor is a clinician-educator [7]. This was also highlighted by the participants (Table 4). The faculty considered the take home message from the three scenario-based discussions to be that mentee should develop his/her own goals, seek help and guidance from mentors and look forward to his/her improvement in all aspects of personal growth and professional practice including career growth. Additionally, faculty mentors also mentioned that there should be a culture of mentorship within the institution from top to bottom level.

## Conclusion and way forward

The process provided valuable training for the development and strengthening of formal mentoring program, provided faculty mentors with clear understanding of their roles and equipped them with the necessary skills required to be an effective faculty mentor role model. It is recommended that institutions should provide the opportunity for development of junior faculty mentors by organizing workshops and other activities so that they get the ability to take up the role effectively in the benefit of the institution, faculty mentor and the faculty mentee. The workshop catered to the needs and perspectives of the faculty mentors at AKU. However, there is a need for identification of the needs of other stakeholders (junior faculty and the institution) and in-depth interviews to strengthen the findings. A formal and structured faculty mentorship model needs to be identified that could be followed in future.

## Limitations

The study was conducted on a small sample size. However, it provides an insight to the faculty mentor understanding of their roles, challenges faced, and the support needed from the institution. The opinion of other stake holders like junior faculty members and institutional leadership should also be considered for a holistic view.

## Supporting information

**S1 Data.**
(SAV)

## Acknowledgments

We are thankful to Dr Fauzia Khan (former chair of FMF for organizing this workshop).

## Author Contributions

**Conceptualization:** Tabassum Zehra, Muhammed Tariq, Rukhsana W. Zuberi.

**Formal analysis:** Tabassum Zehra.

**Methodology:** Tabassum Zehra, Muhammed Tariq.

**Project administration:** Tabassum Zehra.

**Supervision:** Muhammed Tariq.

**Writing – original draft:** Tabassum Zehra.

**Writing – review & editing:** Muhammed Tariq, Rehana Rehman, Rukhsana W. Zuberi.

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
