## [Decision Letter · Decision Letter 0]

19 Dec 2022

PONE-D-22-31892Basics of Faculty-to-Faculty Mentoring: A Reporting Research Process to Identify

Support and ChallengesDear Dr. Zehra,

Thank you for submitting your manuscript to PLOS ONE. After careful consideration, we feel that it has merit but does not fully meet PLOS ONE’s publication criteria as it currently stands. Therefore, we invite you to submit a revised version of the manuscript that addresses the points raised during the review process.

ACADEMIC EDITOR: 

We look forward to receiving your revised manuscript.

Kind regards,

Mukhtiar Baig, Ph.D.

Academic Editor

PLOS ONE

2. In the ethics statement in the Methods, you have specified that verbal consent was obtained. Please provide additional details regarding how this consent was documented and witnessed.

Additional Editor Comments:

The manuscript concept is good, but it needs significant revision.

Reviewers' comments:

Reviewer's Responses to Questions

**Comments to the Author**

1. Is the manuscript technically sound, and do the data support the conclusions?

Reviewer #1: No

Reviewer #2: Partly

2. Has the statistical analysis been performed appropriately and rigorously? 

Reviewer #1: N/A

Reviewer #2: N/A

3. Have the authors made all data underlying the findings in their manuscript fully available?

Reviewer #1: No

Reviewer #2: No

4. Is the manuscript presented in an intelligible fashion and written in standard English?

Reviewer #1: No

Reviewer #2: Yes

5. Review Comments to the Author

Reviewer #1: This study’s objectives are unclear and not mentioned anywhere in the manuscript and thus cannot be correlated with the title. It is unclear whether the objective is to report the process of the workshop or to share the perception of the involved participants about the mentoring program.

The introduction needs major revision as it is not written in a systematic way e.g. there are two places in the first paragraph where there is a mention of a “dearth of literature” but the writing in between these two mentions has no link with them. It should be written in a crisp and organized way.

It is unclear from the manuscript that the intervention is done before the launch of the formal mentoring program in 2019. Not mentioning the timelines gave a sense of confusion in understanding the findings e.g. were the recognized challenges were difficulties that the faculty encountered or potential that they recognized before participating in the program?

The methodology section does not adequately describe the procedure. Even providing an example of a single scenario employed will allow readers to comprehend the process. The duration and timeline of the research process (workshop) should be mentioned when reporting it. It is important to explain how the scenario led to the conclusions and the method utilized to analyze the information from the workshop discussion.

In the result section, the key to the asterisk (*) is not mentioned in table 1.

In terms of its overall structure and case presentation, the manuscript needs significant reworking. It will help other institutions in developing and operating a successful faculty mentoring program if it is improved and presented in a better way.

Reviewer #2: Abstract, introduction, methodology, discussion and conclusion sections are written according to the journal's requirement. Research topic is interesting, and is relevant to medical educators. However, following areas need further elaboration:

1. Research question should be more focused and clear.

2. In the rationale, it is mentioned, "The first workshop was conducted to identify the 1) needs of AKU MC faculty mentors, mentee & institution, ii) explore faculty perspectives on responsibilities & expectations of faculty mentor, faculty mentee, and the institution iii) challenges faced with respect to faculty mentoring iv) scope of mentor-mentee relationship." There are many stakeholders, and needs-assessment cannot be identified with one workshop and with the inclusion of only one stakeholder, the faculty mentors. Other stakeholders, e.g., junior faculty members and institutional leadership should be approached as well to strengthen the findings.

4. If feasible, in-depth qualitative part can be included in methodology to give better interpretations.

5. It is mentioned in the study that formal faculty mentorship program was initiated by the institution, and there is no standard faculty mentorship model to be followed. This point should be elaborated further, as, mostly, when an institution starts a formal mentorship program, it mentions mentorship model in detail with standard operating procedures (SOP).

5. Minor corrections are needed, e.g., full stop (.) is missing in line no. 4 of Abstract after "creating a risk of unexpected outcome", and no. of participants is twenty-two, while, in results section twenty is mentioned.

6. Most of the findings of the research is already present in literature. Authors should highlight additional findings in the local context.

6. PLOS authors have the option to publish the peer review history of their article (what does this mean?). If published, this will include your full peer review and any attached files.

Reviewer #1: No

Reviewer #2: **Yes: **Muhammad Imran

---

## [Author Response · Author response to Decision Letter 0]

7 Feb 2023

First Reviewer’s comment 

This study’s objectives are unclear and not mentioned anywhere in the manuscript and thus cannot be correlated with the title. It is unclear whether the objective is to report the process of the workshop or to share the perception of the involved participants about the mentoring program. Added 

The introduction needs major revision as it is not written in a systematic way e.g. there are two places in the first paragraph where there is a mention of a “dearth of literature” but the writing in between these two mentions has no link with them. It should be written in a crisp and organized way. Revised as suggested

It is unclear from the manuscript that the intervention is done before the launch of the formal mentoring program in 2019. Not mentioning the timelines gave a sense of confusion in understanding the findings e.g. were the recognized challenges were difficulties that the faculty encountered or potential that they recognized before participating in the program? Revised as suggested 

The methodology section does not adequately describe the procedure. Even providing an example of a single scenario employed will allow readers to comprehend the process. The duration and timeline of the research process (workshop) should be mentioned when reporting it. It is important to explain how the scenario led to the conclusions and the method utilized to analyze the information from the workshop discussion. Scenarios added. Revision made

In the result section, the key to the asterisk (*) is not mentioned in table 1. Removed

Second Reviewer’s comments 

Research question should be more focused and clear. Added

There are many stakeholders, and needs-assessment cannot be identified with one workshop and with the inclusion of only one stakeholder, the faculty mentors. Other stakeholders, e.g., junior faculty members and institutional leadership should be approached as well to strengthen the findings. Added in the limitations and way forward 

If feasible, in-depth qualitative part can be included in methodology to give better interpretations. Added in way forward

It is mentioned in the study that formal faculty mentorship program was initiated by the institution, and there is no standard faculty mentorship model to be followed. This point should be elaborated further, as, mostly, when an institution starts a formal mentorship program, it mentions mentorship model in detail with standard operating procedures (SOP). Identified the need for the model and revised 

Minor corrections are needed, e.g., full stop (.) is missing in line no. 4 of Abstract after "creating a risk of unexpected outcome", and no. of participants is twenty-two, while, in results section twenty is mentioned.

 Done

Most of the findings of the research is already present in literature. Authors should highlight additional findings in the local context.

 Done

---

## [Decision Letter · Decision Letter 1]

2 Mar 2023

PONE-D-22-31892R1Basics of Faculty-to-Faculty Mentoring: A Reporting Research to IdentifySupport and ChallengesPLOS ONE

Dear Dr. Zehra,

Thank you for submitting your manuscript to PLOS ONE. After careful consideration, we feel that it has merit but does not fully meet PLOS ONE’s publication criteria as it currently stands. Therefore, we invite you to submit a revised version of the manuscript that addresses the points raised during the review process.

We look forward to receiving your revised manuscript.

Kind regards,

Mukhtiar Baig, Ph.D.

Academic Editor

PLOS ONE

Reviewers' comments:

Reviewer's Responses to Questions

**Comments to the Author**

1. If the authors have adequately addressed your comments raised in a previous round of review and you feel that this manuscript is now acceptable for publication, you may indicate that here to bypass the “Comments to the Author” section, enter your conflict of interest statement in the “Confidential to Editor” section, and submit your "Accept" recommendation.

Reviewer #1: All comments have been addressed

Reviewer #2: All comments have been addressed

2. Is the manuscript technically sound, and do the data support the conclusions?

Reviewer #1: No

Reviewer #2: Yes

3. Has the statistical analysis been performed appropriately and rigorously? 

Reviewer #1: N/A

Reviewer #2: N/A

4. Have the authors made all data underlying the findings in their manuscript fully available?

Reviewer #1: Yes

Reviewer #2: Yes

5. Is the manuscript presented in an intelligible fashion and written in standard English?

Reviewer #1: No

Reviewer #2: Yes

6. Review Comments to the Author

Reviewer #1: As the research question indicates that the purpose of the study is to report participants' perceptions, the methodology section should now be more detailed in terms of data collection and analysis. How the results are compiled and what was the methods used to reach to the reported results.

Please also clearly state the objective no. 1 of your workshop mentioned in the last paragraph of your introduction section

Reviewer #2: Authors have answered all the comments. Abstract is unstructured and all important details are given.

Introduction is comprehensive and objective is clearly mentioned.

Methods, results and discussion sections are written promptly, and all necessary changes have been taken care of adequately.

7. PLOS authors have the option to publish the peer review history of their article (what does this mean?). If published, this will include your full peer review and any attached files.

Reviewer #1: No

Reviewer #2: **Yes: **Muhammad Imran

---

## [Author Response · Author response to Decision Letter 1]

15 Apr 2023

1. Please ensure that your manuscript meets PLOS ONE's style requirements. 

Answer: This has been taken care of. Font and spacing with language corrections were done 

2. In the ethics statement in the Methods, you have specified that verbal consent was obtained. Please provide additional details regarding how this consent was documented and witnessed

Answer: Further details provided

---

## [Editor Report · Decision Letter 2]

31 May 2023

Basics of Faculty-to-Faculty Mentoring: A Reporting Research to IdentifySupport and Challenges

PONE-D-22-31892R2

Dear Dr. Zehra,

We’re pleased to inform you that your manuscript has been judged scientifically suitable for publication and will be formally accepted for publication once it meets all outstanding technical requirements.

Kind regards,

Mukhtiar Baig, Ph.D.

Academic Editor

PLOS ONE

---

## [Editor Report · Acceptance letter]

5 Jun 2023

PONE-D-22-31892R2 

Basics of Faculty-to-Faculty Mentoring: A Process to Identify Support and Challenges. 

Dear Dr. Zehra:

I'm pleased to inform you that your manuscript has been deemed suitable for publication in PLOS ONE. Congratulations! Your manuscript is now with our production department. 

Kind regards, 

on behalf of

Professor Mukhtiar Baig 

Academic Editor

PLOS ONE